# Changes in T-Cell Subpopulations and Cytokine Levels in Patients with Treatment-Resistant Depression—A Preliminary Study

**DOI:** 10.3390/ijms24010479

**Published:** 2022-12-28

**Authors:** Łukasz Piotr Szałach, Wiesław Jerzy Cubała, Katarzyna Aleksandra Lisowska

**Affiliations:** 1Department of Pathophysiology, Faculty of Medicine, Medical University of Gdańsk, 80-210 Gdańsk, Poland; 2Department of Psychiatry, Faculty of Medicine, Medical University of Gdańsk, 80-210 Gdańsk, Poland

**Keywords:** treatment-resistant depression, T cells, IL-8, cytokines, flow cytometry

## Abstract

Although there is some evidence for the involvement of cytokines and T cells in the pathophysiology of treatment-resistant depression (TRD), the nature of this relationship is not entirely clear. Therefore, we compared T-cell subpopulations and serum cytokine levels in TRD patients to find relationships between their immunological profiles, clinical presentation, and episode severity. Blood samples from TRD patients (n = 20) and healthy people (n = 13) were collected and analyzed by flow cytometry. We analyzed the percentages of helper and cytotoxic T cells according to the expression of selected activation markers, including CD28, CD69, CD25, CD95, and HLA-DR. The serum levels of inflammatory cytokines IL12p70, TNF-α, IL-10, IL-6, IL-1β, and IL-8 were also determined. TRD patients had a lower percentage of CD3^+^CD4^+^CD25^+^ and CD3^+^CD8^+^CD95^+^ cells than healthy people. They also had lower serum levels of IL-12p70 and TNF-α, whereas IL-8 levels were significantly higher. Receiver operating characteristic (ROC) analysis demonstrated that serum IL-8 values above 19.55 pg/mL were associated with a 10.26 likelihood ratio of developing TRD. No connections were found between the MADRS score and immunological parameters. These results show that TRD patients have reduced percentages of T cells expressing activation antigens (CD25 and CD95) and higher serum concentrations of proinflammatory and chemotactic IL-8. These changes may indicate reduced activity of the immune system and the important role of IL-8 in maintaining chronic inflammation in the course of depression.

## 1. Introduction

Depression is a severe mental condition that may occur as major depressive disorder (MDD) or bipolar disorder (BD). According to psychoneuroimmunology, mutual interactions between the nervous, endocrine, and immune systems could influence a patient’s mental state [1]. Therefore, dysregulation of immune system activity and chronic inflammation seems to play an essential role in the pathogenesis of depression. 

Several meta-analyses confirmed that MDD patients have increased serum levels of cytokines regulating inflammation, including interleukin (IL)-6, tumor necrosis factor-alpha (TNF-α) [2,3,4], IL-1β, IL-8, IL-10, and IL-18 [5,6,7]. A few studies demonstrated that patients suffering from depression have a reduced percentage of blood lymphocytes [8,9] but a high concentration of serum IL-2, a cytokine essential for T-cell differentiation into effector cells [6,7,10]. Some authors suggest that the stimulating effect of IL-2 is suppressed by the soluble IL-2 receptor (sIL-2R) in the blood [3]. In some other studies, the percentage of CD3^+^ T cells, especially CD3^+^CD4^+^ (helper) T cells, was reported to be increased in MDD patients, whereas the percentage of CD3^+^CD8^+^ (cytotoxic) T cells was reported to be decreased [11,12]. In addition, in meta-analysis, Zorrilla et al. [13] reported an increase in the CD4^+^/CD8^+^ ratio in MDD patients. However, the most recent studies show no evidence that the percentage of T cells or their two central populations, CD4^+^ and CD8^+^ cells, changes during depression [10,14,15]. T cells of MDD patients have also been shown to express some activation markers, including CD95, which is involved in the process of apoptosis [16,17]; CD69, an early activation marker [18,19]; and CD25, an alpha chain of the IL-2 receptor [18,20].

Several studies showed that MDD patients present with high concentrations of IL-12 [10,21], which is involved in promoting Th1-type immune responses and stimulating the production of high levels of interferon-gamma (IFN-γ) [10,22]. Th17 cell number, a subset of proinflammatory T-helper cells, also increases in depression [23,24], most likely due to higher serum IL-6 levels. MDD patients also are characterized by a decreased percentage of CD4^+^CD25^+^FoxP3^+^ cells, which are regulatory T cells (Tregs) responsible for maintaining tolerance against self-antigens, thus preventing autoimmune reactions [9,23,25].

In BD patients, several cytokines are elevated, indicating the hyperactivity of innate immunity. A meta-analysis by Modabbernia et al. [26] showed that serum IL-4, IL-10, TNF-α, sIL-2R, sIL-6R, and soluble TNF receptor 1 (sTNFR1) in BD patients are, in general, significantly increased compared to healthy people. A recent meta-analysis also showed that IL-8 levels in BD patients are significantly increased [27]. Few alterations in the lymphocyte profile in patients with BD have been demonstrated, especially in a subpopulation of T cells. For example, female euthymic patients are characterized by fewer Tregs and a higher number of cytotoxic CD8^+^CD28^−^ T cells than healthy people [28]. Younger euthymic BD patients (younger than 40 years old) have a higher percentage of Tregs [29]. In another study, euthymic BD patients showed significantly increased numbers of Th17 and Tregs compared with healthy controls [30]. A reduced number of cytotoxic T cells and an increased number of activated (CD4^+^CD25^+^) helper T cells with fewer IL-10-expressing Tregs were also described [31].

Although several pharmacological and non-pharmacological therapeutic options are available, treatment resistance among MDD patients is high and can reach up to one-third of patients suffering from depression [32]. Treatment resistance is an inadequate response to two or more antidepressant trials of adequate doses and duration [33]. For bipolar depression, we define treatment-resistant depression (TRD) as a clinically unsatisfactory response following at least two trials of different medicinal treatments in presumably adequate doses and durations within a specific phase of bipolar disorder [34].

Several studies have shown that MDD patients with TRD have higher serum IL-6, IL-8, TNF-α [35], and sIL-6R [36,37] levels than non-TRD patients. However, the most recent study showed that MDD patients with TRD were characterized by lower elevation of IL-6 and TNF-α than non-TRD patients [38]. There are no data available on T-cell changes in this particular patient group. Another study showed that BD patients with TRD were characterized by higher levels of IL-6 and TNF-α and lower levels of sTNFR1 than MDD patients [39]. In BD patients, elevated IL-1β was proposed to predict treatment resistance [40].

In this paper, we compare main T-cell subpopulations and serum cytokine levels in TRD patients to find relationships between patients’ immunological profiles, clinical presentation, and episodic severity. Although there is some evidence for the involvement of cytokines and T cells in TRD pathophysiology, the nature of this relationship is not entirely clear. Therefore, we analyzed the percentages of helper and cytotoxic T cells according to the expression of the CD28 antigen due to its role in antigen presentation and selected activation markers, including CD69, CD25, CD95, and HLA-DR. The serum levels of inflammation-related cytokines IL12p70, TNF-α, IL-10, IL-6, IL-1β, and IL-8 were also determined.

## 2. Results

### 2.1. Comparison of T-Cell Subpopulations and Serum Cytokine Levels between TRD Patients and Healthy People

The study groups consisted of 20 inpatients diagnosed with TRD in the course of MDD or BD without psychotic features and 13 healthy people (Table 1). There was no difference between TRD patients and healthy people in terms of age or BMI.

The percentage of CD3^+^ T cells and their main subpopulations, CD3^+^CD4^+^ cells (helper T cells) and CD3^+^CD8^+^ (cytotoxic T cells) cells, were analyzed in healthy people and TRD patients, then compared with each other. In addition, in each of the T-cell populations, we analyzed the percentage of CD28^+^ (costimulatory antigen), CD69^+^ (early-activation antigen), CD25^+^, HLA-DR^+^ (middle-activation antigens), and CD95^+^ (activation protein also involved in apoptosis) subpopulations. Figure 1 shows an exemplary cytometric analysis of T-cell subpopulations in TRD patients.

We found no significant difference in the central populations of T cells (CD3^+^, CD3^+^CD4^+^, and CD3^+^CD8^+^ cells) between healthy people and TRD patients (Table 2). There were differences in the percentage of CD3^+^CD4^+^CD25^+^ (Figure 1D,E) and CD3^+^CD8^+^CD95^+^ cells. TRD patients had a lower percentage of CD3^+^CD4^+^CD25^+^ cells (median value of 6.99 compared to 13.16 in HC) (Table 2, Figure 2A). They also had a lower percentage of CD3^+^CD8^+^CD95^+^ cells (median value of 18.2 compared to 30.3 in HC) (Table 2, Figure 2B). No significant changes in other examined subpopulations were found between groups (Table 2).

There were significant differences in serum cytokine levels between TRD patients and healthy people (Table 2). Healthy people had a higher level of serum IL-12p70 (median value of 2.65 pg/mL) compared to TRD patients, who, in most cases, had IL-12p70 contents below the level of detection (median value of 0 pg/mL) (Table 2, Figure 3A). They also had higher TNF-α level (median value of 1.2 pg/mL) compared to TRD patients (median value of 0 pg/mL) (Table 2, Figure 3B). 

On the other hand, IL-8 levels were significantly increased in TRD patients (median value of 185.9 pg/mL) compared to healthy people (median value of 11.33 pg/mL) (Table 2, Figure 3C). Receiver operating characteristic (ROC) analysis demonstrated that serum IL-8 levels above 19.55 pg/mL were associated with a 10.26 likelihood ratio of developing TRD, with a sensitivity of 78.95% and a specificity of 92.31% (Figure 4).

### 2.2. Correlations between Immunological and Clinical Parameters

A psychiatrist assessed all patients with the MADRS scale. The median MADRS score in MDD patients was 32.5. We performed a correlation test of clinical characteristics and immunological parameters in both patient groups. No correlation was observed between MADRS and any immunological parameter in TRD patients. MADRS was only negatively correlated with patient BMI (r = −0.563, *p* = 0.010). Patient age was also negatively correlated with serum IL-1β (r = −0.548, *p* = 0.015) and IL-8 (r = −0.502, *p* = 0.029).

Additionally, a positive correlation was found between the length of the current episode and percentage of CD3^+^CD4^+^ cells (r = 0.620, *p* = 0.004), and a negative correlation was identified between episode length and percentage of CD3^+^CD8^+^ cells (r = −0.599, *p* = 0.005) (Table 3). Current episode length was also positively correlated with the CD4^+^/CD8^+^ ratio (r = 0.604, *p* = 0.005). After adjusting for potential confounders (sex, age, BMI, and smoking), there was a significant interaction between these variables.

### 2.3. Correlations between Immunological Parameters

We have found some correlations between the percentages of T-cell subpopulations and serum cytokine levels in TRD patients (Table 4). There was a significant positive correlation between IL-10 and CD3^+^CD8^+^ (r = 0.468, *p* = 0.043) and CD3^+^CD8^+^CD95^+^ cells (r = 0.579, *p* = 0.021). There was a negative correlation between IL-10 and CD3^+^CD4^+^ cells (r = −0.473, *p* = 0.041) and between IL-10 and the CD4^+^/CD8^+^ ratio (r = −0.470, *p* = 0.042). There was a positive correlation between IL-6 and CD3^+^CD8^+^CD95^+^ cells (r = 0.573, *p* = 0.022). IL-1β concentration was positively correlated with CD3^+^CD4^+^CD25^+^ (r = 0.573, *p* = 0.01) and CD3^+^CD4^+^CD69^+^ (r = 0.592, *p* = 0.014) cells. IL-8 levels correlated positively with CD3^+^CD4^+^CD69^+^ cells (r = 0.564, *p* = 0.02) and CD3^+^CD4^+^HLA-DR^+^ (r = 0.544, *p* = 0.026) cells.

## 3. Discussion

Our results show several significant findings in the immune profiles of TRD patients. First, they had significantly higher serum IL-8 levels, with a median of 185.9 pg/mL, which is sixteen times higher compared to healthy people. Additionally, TRD patients were characterized by lower serum IL-12p70 and TNF-α levels than healthy people. Third, we found no statistically significant differences in the most often described proinflammatory cytokines, such as IL-1β and IL-6. Finally, with respect to T-cell subpopulations, the median percentage values of CD3^+^CD4^+^CD25^+^ and CD3^+^CD8^+^CD95^+^ cells were twice as low as in healthy people, without any changes in other subpopulations. 

As mentioned in the Introduction, some studies have shown that TRD patients have higher serum levels of IL-6 and TNF-α than non-TRD patients [35,39]. However, other studies show that TRD patients are characterized by a lower elevation of IL-6 and TNF-α compared to non-TRD patients [38]. Kiraly et al. [41] recently demonstrated that TRD patients have only high serum IL-6 compared with healthy people, without any changes in IL-1β or TNF-α. In our study, TRD patients had lower TNF-α levels, without changes in IL-1β or IL-6. TNF-α is a proinflammatory cytokine produced by several cells, primarily macrophages, in response to bacterial products and IL-1 signaling [18]. TNF-α, IL-1β, and IL-6 are mainly involved in acute inflammation, leading to healing, trigger removal, tissue repair, and some systemic autoimmune diseases such as rheumatoid arthritis. Chronic inflammation accompanying some diseases, including depression, is characterized by a lack of canonical biomarkers [42]. The diagnostic value of IL-1β or TNF-α is also somewhat limited due to their half-life of 5–10 min [43]. In our study, the serum level of IL-1β was slightly decreased in TRD patients, which could explain the lower TNF-α level observed in patient sera. Lower (sometimes even undetectable) cytokine levels in TRD patients could indicate reduced macrophage activation due to prolonged, chronic inflammation.

IL-12p70, which is mainly produced by macrophages and dendritic cells, induces activation and proliferation of Th1 cells, which are responsible for cell-mediated response to pathogens [44]. According to some authors, MDD patients present with high concentrations of IL-12 [10,21]. In our study, TRD patients were characterized by lower serum IL-12p70 levels than healthy people, which may indicate a shift in immune response from Th1 (cellular) to Th2 (humoral) response. Similar results were reported by Cosma et al. [45]. However, in vitro studies demonstrated increased activation of M1 macrophages, which are responsible for activating Th1 responses. This difference may occur because in circulation, cells are influenced by hormones and neurotransmitters, including glucocorticoids, whereas in vitro, this influence is limited.

We also observed very high IL-8 levels in TRD patients, with a maximum of 2156 pg/mL, compared to a maximum level of 20.2 pg/mL in healthy people. Similar results were reported by Strawbridge et al. [35], who also suggested that elevated IL-8 is indicative of treatment-resistant MDD. Using ROC analysis, we were able to confirm that high serum IL-8 levels are associated with a 10.26 likelihood ratio of developing TRD. IL-8 is a crucial mediator that is mainly produced by monocytes, macrophages, and neutrophils and mainly responsible for neutrophil migration. However, its role in depression is currently unclear. Some previous findings indicated that human microglia can synthesize IL-8 in response to proinflammatory stimuli and that anti-inflammatory cytokines downregulate the production of this chemokine [46]. Furthermore, in a study by Taub et al. [47], a marked T-cell infiltrate was observed in the IL-8 skin injection site, confirming that IL-8 is also a chemoattractant for lymphocytes.

We failed to demonstrate any correlation between cytokines and MADRS score. The latest study by Kofod et al. [48] showed no correlation between any measured inflammatory markers, including IL-1β, IL-6, TNF-α, or IL-8, and MADRS in patients with moderate–severe depression. In our study, the MADRS score was negatively correlated only with patient BMI. Obesity can be characterized by a chronic low-grade inflammatory state that originates primarily from adipocytes capable of producing proinflammatory cytokines. In a study by Huet et al. [49], severely obese MDD patients (BMI > 40 kg/m^2^) had higher MADRS scores compared to lean people and people with overweight or moderate obesity. However, in another study, no relationship was between MADRS and the severity of obesity measured by BMI, although women with low BMI (≤18.5 kg/m^2^) achieved significantly higher MADRS scores than other women [50]. Differences in body fat distribution among TRD patients could explain disparities in these results. Whereas BMI is the most common measure of obesity, it is not fully representative of body fat distribution, especially central adiposity, because, compared to subcutaneous adipose tissue, visceral adipose tissue is considered to be associated with inflammation [51]. 

We found no difference in the percentage of T cells or their two central populations, CD3^+^CD4^+^ (helper) and CD3^+^CD8^+^ (cytotoxic) cells, which is in line with reports by other authors [9,14,15]. However, we observed correlations between current episode length and main T-cell subpopulations: a positive correlation with the percentage of CD3^+^CD4^+^ and the CD4^+^/CD8^+^ ratio and negative correlation with CD3^+^CD8^+^ cells. The results obtained in this study suggest that the longer the current episode, the higher the percentages of helper T cells. However, because no similar results have been reported in the literature, the meaning of this correlation remains unclear. We also observed correlations between percentages of several T-cell subpopulations and serum levels of IL-10, IL-6, IL-1β, and IL-8, confirming that the lymphocyte profile in TRD patients is related to inflammation-related cytokines. Proinflammatory IL-1β and IL-6 were mainly correlated with T cells expressing activation antigens (CD25, CD69, HLA-DR, or CD95). Moreover, anti-inflammatory IL-10 was negatively correlated with CD3^+^CD4^+^ cells, possibly as a response to chronic inflammation. 

The median percentage of CD3^+^CD4^+^CD25^+^ cells in TRD patients was nearly twice as low as that in healthy people. This finding is consistent with previous studies on lymphocyte profiles in MDD patients [2,12,22], where CD25-positive cells were identified as Tregs. Unfortunately, the lack of a Treg marker, i.e., the FoxP3 transcription factor, does not allow us to indisputably conclude that we are dealing with a Treg deficiency. The CD25 antigen is also an activation marker in CD4^+^ and CD8^+^ T cells. Therefore, CD25-positive cell deficiency may be associated with an inadequate T-cell response as expressed by changes in serum IL-12p70 or TNF-α in TRD patients.

Previous studies reported an association between increased expression of CD95 on T cells with increased cell susceptibility to apoptosis [16,48]. In our study, TRD patients were characterized by lower percentages of CD3^+^CD8^+^CD95^+^ cells. On the other hand, no change was found in the percentage of CD3^+^CD4^+^CD95^+^ cells. Unfortunately, the lack of newer studies makes it impossible to compare the obtained results. However, because CD95 is also associated with the activation of T cells, its deficiency, together with the deficiency of CD25-positive cells, could be associated with disorders of T-cell activation in TRD patients. 

We did not observe differences in other antigens involved in T-cell activation, CD69, or HLA-DR. The CD69 antigen is a classical early activation marker that appears an hour after T-cell contact with the antigen and rapidly decreases within 4–6 h. In chronic conditions, CD69 is expressed on infiltrated leukocytes at inflammatory sites rather than in blood [52], which could explain why we did not observed changes in its expression in T cells from blood in TRD patients. HLA-DR, on the other hand, is a molecule typically expressed by antigen-presenting cells and is associated with antigen presentation. Therefore, this antigen is also expressed in T cells during their activation. According to some authors, reduced HLA-DR expression is a mechanism that adapts to an overwhelming burst of inflammatory stimuli; a high frequency of HLA-DR^+^ T cells strongly correlates with severe lymphopenia, systemic inflammation, and cytokine storm rather than with chronic conditions [53].

The present study is subject to potential limitations. First, the included group samples are relatively small. Hence, we were not able to separate several subgroups based on the factors that might have influenced the immune system, such as specific psychiatric medications. The limited number of patients is particularly problematic in studying the T-cell phenotype in TRD. Second, bipolar depression may have been misdiagnosed as MDD because depressive episodes during BD might be indistinguishable from unipolar depression. Third, we did not compare TRD patients with non-TRD patients. Furthermore, our observations and those of other authors show that serum cytokine levels in depressed patients vary, which may be related to the type of treatment and the response to this treatment. Fifth, our study included only Caucasian individuals in each of the study groups. Therefore, our results might not be fully applicable to different ethnic groups.

## 4. Materials and Methods

### 4.1. Study Groups

The study groups consisted of 20 inpatients diagnosed with depression in the course of MDD or BD without psychotic features (Table 1). Subjects were examined by clinicians using the Mini International Neuropsychiatric Interview (MINI) to verify the diagnosis using the Diagnostic and Statistical Manual of Mental Disorders (DSM-5) criteria, as well as with the Montgomery–Asberg Depression Rating Scale (MADRS). In addition, all participants were described as treatment-resistance during the current episode, with MDD patients assessed by the Massachusetts General Hospital Antidepressant Treatment Response Questionnaire. A total of 13 healthy people of similar age were also included in the study.

The exclusion criteria for the present study were uncontrolled arterial hypertension; unstable coronary artery disease; increased intracranial pressure; acute and chronic infectious diseases; inflammatory, autoimmune, and metabolic diseases; and neoplastic diseases. An additional inclusion criterion for the control group was an absence of mental disorders. Only medically stable adult inpatients who were able to communicate and provide consent were enrolled in the study. All subjects provided written informed consent to participate in the study. For each participant, written consent was obtained after the procedures had been fully explained. The Independent Bioethics Committee for Scientific Research approved the study (consent nos. NKBBN/172/2017 and NKBBN/172-674/2019). A flow chart of the study methodology is presented in Figure 5.

The tested material was 5 mL of peripheral venous blood collected in EDTA tubes to determine lymphocyte subpopulations. In addition, 5 mL of blood was collected in anticoagulant-free tubes to collect serum to assess concentrations of cytokines. We stored serum samples at 80 °C.

### 4.2. Ex Vivo Determination of T-Cell Subpopulations 

Samples of 100 μL per tube blood were transferred for staining with monoclonal antibodies and red blood cell (RBC) lysis. RBCs were lysed with buffer containing 0.8% NH_4_Cl and 0.1% KHCO_3_. Cells were then washed with PBS (phosphate-buffered saline) and stained with FITC-conjugated anti-CD3 or anti-CD95, RPE-Cy5-conjugated anti-CD4 (BD Pharmingen, San Diego, CA, USA), APC-H7-conjugated anti-CD8, PE-conjugated anti-CD28, anti-CD25, anti-CD69, or anti-HLA-DR (BD Pharmingen, USA) for 30 min at 4 °C in the dark. Then, cells were washed with PBS and suspended in 200 mL of suitable buffer for flow cytometric analysis using a FACSVerse instrument (Becton Dickinson, Franklin Lakes, NJ, USA).

### 4.3. Cytokine Measurement in Plasma Samples

A cytometric bead array (CBA) human inflammatory cytokine kit (BD Biosciences, USA) was used according to the manufacturer’s protocol to determine the levels of IL12p70, TNF-α, IL-10, IL-6, IL-1β, and IL-8 in serum samples from healthy people and both groups of patients. Quantitative cytometric fluorescence analysis was performed using a FACScan cytometer (Becton Dickinson, USA). The detection range for all measured cytokines was between 20 and 5000 pg/mL. The kit performance was optimized to analyze physiologically relevant concentrations (pg/mL levels) of specific cytokine proteins in tissue culture supernatants and serum samples. The limit of detection range for IL-8 was 3.6 pg/mL, 7.2 pg/mL for IL-1β, 2.5 pg/mL for IL-6, 3.3 pg/mL for IL-10, 3.7 pg/mL for TNF-α, and 1.9 pg/mL for IL-12p70. 

### 4.4. Analysis and Statistics

Twenty thousand events corresponding to the light-scatter characteristics of viable lymphocytes were acquired from each sample to analyze T-cell subpopulations. First, lymphocytes were selected based on forward- and side-scatter characteristics (Figure 1A). Then, T cells were identified based on their positivity for the CD3 antigen (Figure 1B). Next, helper T cells were identified based on the expression of the CD4 antigen, and cytotoxic T cells were identified based on CD8 expression (Figure 1C). Finally, subpopulations expressing different activation antigens, e.g., the CD25 antigen (Figure 1D,E), were identified. Cytometric data were analyzed with FlowJo 10 software (Beckton Dickinson, Franklin Lakes, NJ, USA).

Statistical data analysis was conducted using GraphPad 9 (version 9) statistical software. First, the distribution of the examined variables was checked with Kolmogorov–Smirnov and Shapiro–Wilk normality tests. Then, depending on data distribution, quantitative variables were analyzed with a two-tailed unpaired *t*-test, Mann–Whitney U test, or Spearman rank correlation test. A significance level of *p* < 0.05 was set for all analyses.

## 5. Conclusions

Although there is some evidence for the involvement of cytokines and T cells in TRD pathophysiology, the nature of this relationship is not entirely clear. Our results show that TRD patients have reduced percentages of T cells expressing activation antigens (CD25 and CD95) and higher serum levels of proinflammatory and chemotactic IL-8. These changes may indicate reduced activity of the immune system and the important role of IL-8 in maintaining chronic inflammation in the course of depression. Analyzing immunological parameters in these groups of patients requires larger sample sizes. The human immune system is susceptible to environmental factors and has many connections with the endocrine and nervous systems. Large study groups or prospective studies may answer the problem of differential observations in these groups of patients, making it possible to study the complicated relationships between TRD and response to treatment or correlation with disease activity.

## Figures and Tables

**Figure 1 ijms-24-00479-f001:**
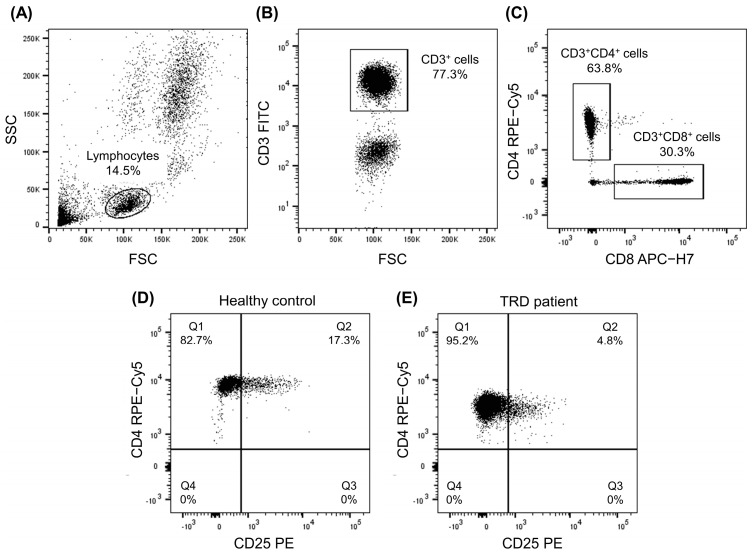
Cytometric analysis of T-cell subpopulations. Lymphocytes were selected based on forward- and side-scatter characteristics (**A**). Then, T cells were identified based on their positivity for the CD3 antigen (**B**). Next, helper T cells were identified based on the expression of the CD4 antigen and cytotoxic T cells based on CD8 expression (**C**). Finally, subpopulations expressing different activation antigens, e.g., the CD25 antigen, were identified. Exemplary dot plots with CD4^+^CD25^+^ cells from a healthy donor (**D**) and from a TRD patient (**E**).

**Figure 2 ijms-24-00479-f002:**
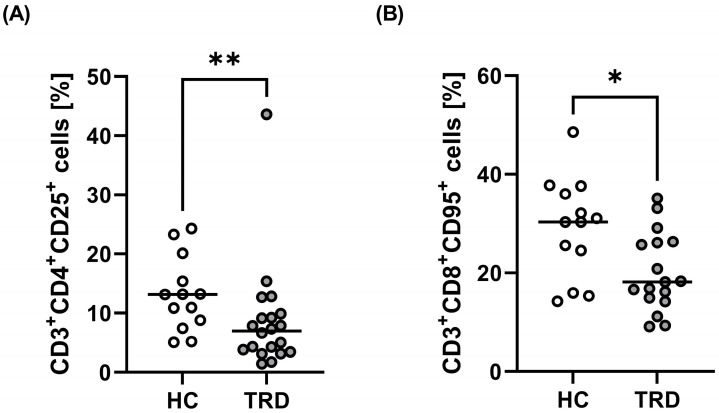
Comparison of percentage of CD4^+^CD25^+^ cells (**A**) and CD3^+^CD8^+^CD95^+^ cells (**B**) in TRD patients and healthy people. The line represents the median according to two-tailed Mann–Whitney U test; * *p* < 0.05, ** *p* < 0.01.

**Figure 3 ijms-24-00479-f003:**
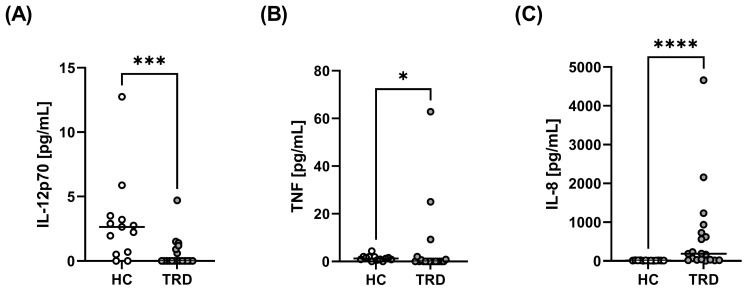
Comparison of serum concentrations of IL-12p70 (**A**), TNF-α (**B**), and IL-8 (**C**) between TRD patients and healthy people. The line represents the median according to two-tailed Mann–Whitney U test; * *p* < 0.05, *** *p* < 0.001, **** *p* < 0.0001.

**Figure 4 ijms-24-00479-f004:**
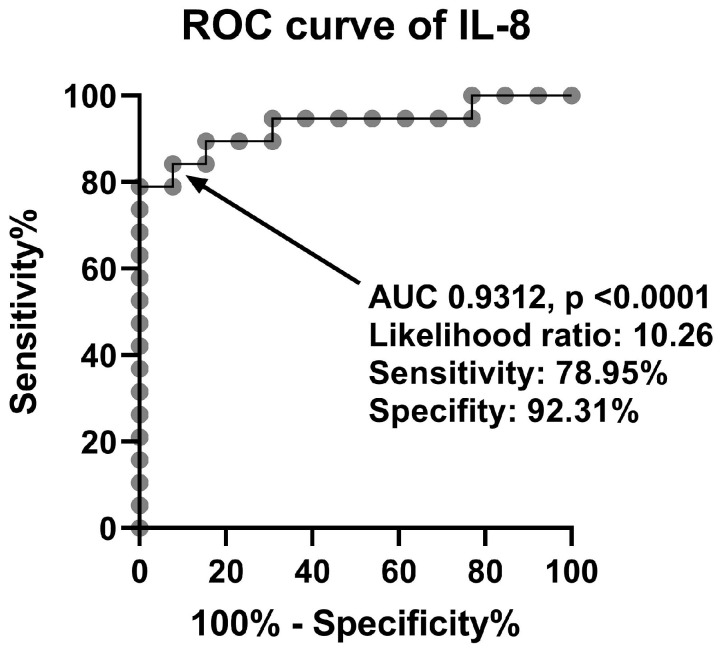
ROC curve for IL-8 in TRD patients.

**Figure 5 ijms-24-00479-f005:**
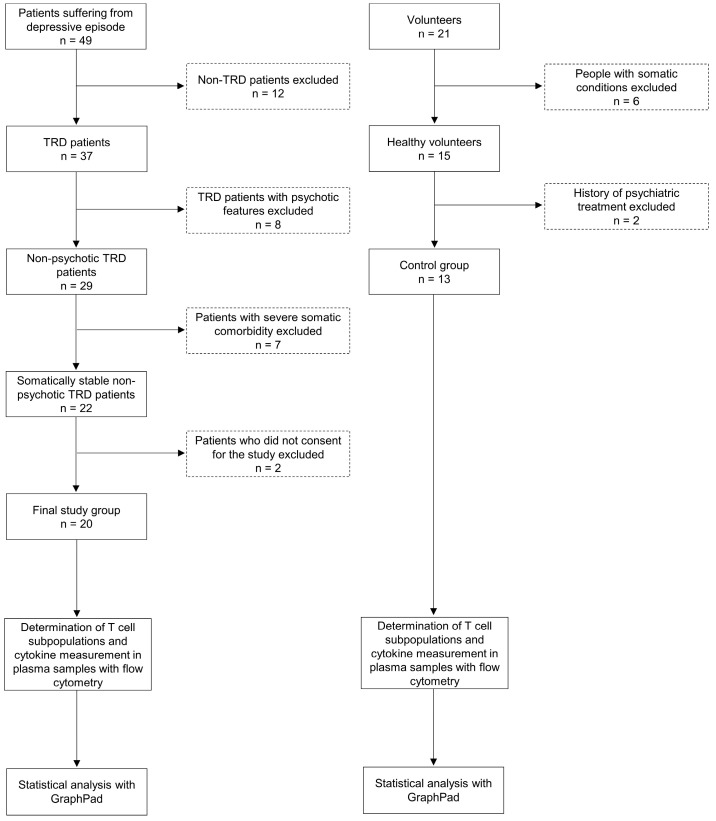
Flow chart of the study methodology.

**Table 1 ijms-24-00479-t001:** Basic characteristics of study participants.

	HC (n = 13)	TRD (n = 20)	*p* Value
Sex (male/female)	7/6	9/11	0.728
Age (years)	51 ± 17.63	47.50 ± 19.03	0.509
BMI (kg/m^2^)	24.47 ± 5.34	28.24 ± 6.27	0.115
Length of current episode (weeks)	n.a.	38 ± 6.08	n.a.
MADRS	n.a.	30.2 ± 1.49	n.a.
Medication *			
SSRIs	n.a.	5	n.a.
SNRIs	n.a.	7	n.a.
SARIs	n.a.	2	n.a.
NaSSAs	n.a.	6	n.a.
TCAs	n.a.	2	n.a.
Atypical antipsychotics	n.a.	10	n.a.
Antiepileptic drugs	n.a.	5	n.a.
Low-potency antipsychotics	n.a.	4	n.a.
Lithium	n.a.	6	n.a.

Data are shown as mean results with standard deviation according to unpaired *t*-test. HC—healthy control, TRD—treatment-resistant depression, BMI—body mass index, SSRIs—serotonin reuptake inhibitors, SNRIs—serotonin and norepinephrine reuptake inhibitors, SARIs—serotonin antagonist and reuptake inhibitors, NaSSAs—noradrenergic and specific serotonergic antidepressants, TCAs—tricyclic antidepressants. * Please note that some patients received more than one medication.

**Table 2 ijms-24-00479-t002:** Comparison of T-cell subpopulations and cytokine levels between TRD patients and healthy people.

	HC	TRD	*p* Value
CD3^+^ (%)	75.66 (62.2; 80.82)	74.35 (43.3; 83.6)	0.407
CD3^+^CD4^+^ cells (%)	61.4 (44.54; 76.88)	67 (48.8; 82.5)	0.286
CD3^+^CD8^+^ cells (%)	32.8 (17.01; 48)	32.65 (13.2; 48.5)	0.630
CD4^+^/CD8^+^ ratio	1.95 (1.08; 3.95)	2.05 (1.03; 6.25)	0.964
CD3^+^CD4^+^ CD28^+^ cells (%)	59.02 (38.98; 69.81)	60 (36.7; 82.1)	0.434
CD3^+^CD4^+^CD25^+^ cells (%)	**13.16 (5.08; 24.3)**	**6.99 (1.42; 43.6)**	**0.006**
CD3^+^CD4^+^CD69^+^ cells (%)	0.63 (0.04; 12.5)	0.4 (0.03; 20)	0.242
CD3^+^CD4^+^CD95^+^ cells (%)	35.8 (18.3; 53.33)	29.1 (17; 45.8)	0.080
CD3^+^CD4^+^HLA-DR^+^ cells (%)	1.68 (0.44; 7.18)	1.11 (0.09; 20.5)	0.715
CD3^+^CD8^+^CD28^+^ cells (%)	17.71 (9.59; 28.1)	18.95 (11.2; 35.3)	0.377
CD3^+^CD8^+^CD25^+^ cells (%)	0.64 (0.06; 4.84)	0.39 (0.06; 3.82)	0.191
CD3^+^CD8^+^CD69^+^ cells (%)	0.44 (0.07; 2.92)	0.45 (0.01; 2.44)	0.805
CD3^+^CD8^+^CD95^+^ cells (%)	**30.3 (14.23; 48.56)**	**18.2 (9.11; 35.1)**	**0.024**
CD3^+^CD8^+^HLA-DR^+^ cells (%)	2.79 (0.92; 13.5)	8.8 (0.57; 15.9)	0.059
IL-12p70 (pg/mL)	**2.65 (0; 12.74)**	**0 (0; 4.7)**	**0.001**
TNF-α (pg/mL)	**1.2 (0; 4.36)**	**0 (0; 62.8)**	**0.025**
IL-10 (pg/mL)	2.7 (0; 6.15)	1.8 (0; 5.6)	0.153
IL-6 (pg/mL)	2.89 (0; 22.57)	2.8 (0; 110.4)	0.726
IL-1β (pg/mL)	2.26 (0; 18.3)	0.7 (0.0; 12.6)	0.133
IL-8 (pg/mL)	**11.33 (5.5; 20.2)**	**185.9 (8.1; 4660)**	**<0.0001**

Data are presented as median with minimum and maximum results according to two-tailed Mann–Whitney U test. The results in bold are statistically significant at *p* < 0.05.

**Table 3 ijms-24-00479-t003:** Correlations between current episode length and immune parameters.

	r	*p* Value
CD3^+^ (%)	−0.160	0.500
CD3^+^CD4^+^ cells (%)	**0.620**	**0.004**
CD3^+^CD8^+^ cells (%)	**−0.599**	**0.005**
CD4^+^/CD8^+^ ratio	**0.604**	**0.005**
CD3^+^CD4^+^CD28^+^ cells (%)	0.364	0.115
CD3^+^CD4^+^CD25^+^ cells (%)	0.007	0.977
CD3^+^CD4^+^CD69^+^ cells (%)	0.303	0.222
CD3^+^CD4^+^CD95^+^ cells (%)	0.482	0.052
CD3^+^CD4^+^HLA-DR^+^ cells (%)	0.230	0.359
CD3^+^CD8^+^ CD28^+^ cells (%)	−0.358	0.121
CD3^+^CD8^+^CD25^+^ cells (%)	−0.052	0.826
CD3^+^CD8^+^CD69^+^ cells (%)	0.040	0.875
CD3^+^CD8^+^CD95^+^ cells (%)	−0.249	0.332
CD3^+^CD8^+^HLA-DR^+^ cells (%)	−0.102	0,686
IL-12p70 (pg/mL)	0.208	0.394
TNF-α (pg/mL)	0.127	0.605
IL-10 (pg/mL)	−0.273	0.259
IL-6 (pg/mL)	0.142	0.562
IL-1β (pg/mL)	0.058	0.815
IL-8 (pg/mL)	−0.031	0.901

Two-tailed Spearman rank correlation test. The results in bold are statistically significant at *p* < 0.05.

**Table 4 ijms-24-00479-t004:** Correlations between immunological parameters in TRD patients.

	IL-12p70 (pg/mL)	TNF-α (pg/mL)	IL-10 (pg/mL)	IL-6 (pg/mL)	IL-1β (pg/mL)	IL-8 (pg/mL)
CD3^+^ (%)	−0.195	−0.233	0.218	−0.045	0.061	−0.168
CD3^+^CD4^+^ cells (%)	−0.006	0.049	**−0.473**	0.091	0.072	0.226
CD3^+^CD8^+^ cells (%)	0.040	−0.03	**0.468**	−0.087	−0.037	−0.242
CD4^+^/CD8^+^ ratio	−0.023	0.039	**−0.470**	0.092	0.054	0.234
CD3^+^CD4^+^CD28^+^ cells (%)	0.168	−0.089	−0.312	0.133	0.214	0.135
CD3^+^CD4^+^CD25^+^ cells (%)	0.396	0.013	0.236	0.258	**0.573**	0.363
CD3^+^CD4^+^CD69^+^ cells (%)	0.401	0.470	0.046	0.395	**0.592**	**0.564**
CD3^+^CD4^+^CD95^+^ cells (%)	−0.064	−0.029	−0.304	0.272	0.139	−0.085
CD3^+^CD4^+^HLA-DR^+^ cells (%)	−0.066	0.119	−0.378	0.219	0.416	**0.544**
CD3^+^CD8^+^CD28^+^ cells (%)	0.295	−0.056	0.190	−0.172	0.236	−0.188
CD3^+^CD8^+^CD25^+^ cells (%)	0.336	−0.018	0.242	0.236	0.301	−0.182
CD3^+^CD8^+^CD69^+^ cells (%)	−0.104	0.040	0.165	0.088	0.124	0.239
CD3^+^CD8^+^CD95^+^ cells (%)	0.224	0.132	**0.579**	**0.573**	0.327	0.003
CD3^+^CD8^+^HLA-DR^+^ cells (%)	0.075	−0.072	−0.183	−0.022	0.248	0.088

Two-tailed Spearman rank correlation test. The results in bold are statistically significant at *p* < 0.05.

## Data Availability

The data presented in this study are available upon request from the corresponding author.

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
