# Peer review of "Changes in T-Cell Subpopulations and Cytokine Levels in Patients with Treatment-Resistant Depression—A Preliminary Study"

_ijms, 2022, doi:10.3390/ijms24010479_

Round 1
Reviewer 1 Report
Manuscript entitled " Changes in T cell subpopulations and cytokine levels in patients with treatment-resistant depression " is interesting and important, especially since depression has been a serious problem in recent years that affects a large percentage of the population. Despite my overall enthousiasm, I would suggest some revisions. Please see below.
1. The study group is so small, the treatments used are so varied, which makes the results very uncertain. In view of the above, perhaps the title should be supplemented with the term "preliminary study".
2. Is there any possibility that the concentration of cytokines and immunological parameters is due to a certain confounding factor not the presence of the treatment- resistant depression? If you analyze the concentration of cytokines and immunological parameters by adjusting the confounding factors, are some cytokines and immunological parameters still correlated with the presence of depression? eg. gender, hormones treatment, tobacco, obesity, insomnia, etc.?? Especially, that the concentration of IL-1β and IL-8 correlated with age.
3. What may be the reason for such a high concentration of IL-8, if other pro-inflammatory cytokines do not differ or are below the detection level of the ELISA kit in the study group.
4. In the abstract, add the number of patients (n).
5. TNF-α concentration was analyzed in the article? Please specify. Once, you write TNF, another TNF-α.
6. On line: 222 please correct MBI to BMI.
7. I have doubts about the indicated relationship: " MADRS score was negatively correlated only with patients’ BMI”. If the relationship were positive, then the indicated explanation might be important, I am not convinced of this result.
8. Since IL-12p70 and TNF concentrations were below the quantifiability of the ELISA kits, high sensitivity kits should be considered.
9. It is interesting, whether the concentration of the tested cytokines correlated with the immunological parameters? Perhaps, this would explain the better observed lymphocyte profile, and the importance of obtained the results in depression. Especially since not only chronic inflammation, but perhaps, an impaired anti-inflammatory response may underlie depression?
Author Response
Answers to the first Reviewer. Corrections are marked in yellow.
"Manuscript entitled " Changes in T cell subpopulations and cytokine levels in patients with treatment-resistant depression " is interesting and important, especially since depression has been a serious problem in recent years that affects a large percentage of the population. Despite my overall enthousiasm, I would suggest some revisions. Please see below."
Thank you for your comments and suggestions.
"The study group is so small, the treatments used are so varied, which makes the results very uncertain. In view of the above, perhaps the title should be supplemented with the term "preliminary study"."
We agree that the study group is small. Therefore, we followed your suggestion and added the phrase "a preliminary study" in the title.
"Is there any possibility that the concentration of cytokines and immunological parameters is due to a certain confounding factor not the presence of the treatment- resistant depression? If you analyze the concentration of cytokines and immunological parameters by adjusting the confounding factors, are some cytokines and immunological parameters still correlated with the presence of depression? eg. gender, hormones treatment, tobacco, obesity, insomnia, etc.?? Especially, that the concentration of IL-1β and IL-8 correlated with age."
First, although we saw differences in some immune parameters between HC and TRD, we did not find any correlations between MADRS or immune parameters. There was no difference in the serum IL-1β level. We saw only correlations between episode length and a few T cell subpopulations (CD3+CD4+, CD3+CD8+ and their ratios, Table 3). After adjusting for potential confounders (sex, age, BMI, and smoking), there was a significant interaction between these variables. We added this information in the results (page 6).
"What may be the reason for such a high concentration of IL-8, if other pro-inflammatory cytokines do not differ or are below the detection level of the ELISA kit in the study group."
First, IL-8 is not only a pro-inflammatory cytokine but also a chemokine that targets neutrophils, basophils, and T-cells during the inflammatory process. It is not a chemotactic factor for monocytes. IL-8 is involved in neutrophil activation and is released from several cell types in response to inflammation, including monocytes, macrophages, neutrophils, and intestine, kidney, placenta, and bone marrow cells. Its role in chronic inflammatory processes is not precisely clear. Recent studies suggest prolonged and excessive inflammatory stimulation can lead to neutrophil dysregulation and dysfunction, which are significant drivers of innate immune dysfunction and multiorgan dysfunction syndrome during sepsis. Some older findings indicate that human microglia synthesize IL-8 in response to pro-inflammatory stimuli and that anti-inflammatory cytokines down-regulate the production of this chemokine [Ehrlich LC et al. J Immunol. 1998 Feb 15;160(4):1944-8]. A study by [Taub DD et al. J Clin Invest. 1996 Apr 15;97(8):1931-41], a marked human T cell infiltrate was observed in the IL-8 skin injection site. We could not find more recent studies explaining its role in depression. We added this information to the Discussion (page 8).
"In the abstract, add the number of patients (n)."
We added this information in the abstract.
"TNF-α concentration was analyzed in the article? Please specify. Once, you write TNF, another TNF-α."
TNF-α was analyzed. We corrected it in the text.
"On line: 222 please correct MBI to BMI."
We corrected it in the text.
"I have doubts about the indicated relationship: " MADRS score was negatively correlated only with patients' BMI". If the relationship were positive, then the indicated explanation might be important, I am not convinced of this result."
As we wrote in the Discussion, it is known that obesity can be characterized by a chronic low-grade inflammatory state that originates primarily from adipocytes able to produce pro-inflammatory cytokines. There are studies showing that severely obese MDD patients (BMI > 40 kg/m2) had higher MADRS scores compared to lean people or people with overweight or moderate obesity [Huet et al. Brain Behav Immun. 2021 May;94:104-110]. But there are also studies demonstrating no relationship between the MADRS and the severity of obesity measured with BMI [Berlin I et al. Eur Psychiatry. 2003 Mar;18(2):85-8]. Moreover, women with low BMI (≤ 18.5 kg/m2) achieve significantly higher MADRS scores than other women. Differences in body fat distribution among TRD patients could explain disparities in these results. While BMI is the most common measure of obesity, it is not fully representative of body fat distribution, especially central adiposity, because, compared to the subcutaneous adipose tissue, the visceral adipose tissue is considered associated with inflammation [Yu JY et al. Medicine (Baltimore). 2019 Mar;98(9):e14740].
"Since IL-12p70 and TNF concentrations were below the quantifiability of the ELISA kits, high sensitivity kits should be considered."
The kit performance has been optimized to analyze physiologically relevant concentrations (pg/ml levels) of specific cytokine proteins in tissue culture supernatants and serum samples. The limit of detection range for IL-8 was 3.6 pg/ml, for IL-1β – 7.2 pg/ml, IL-6 – 2.5 pg/ml, IL-10 – 3.3 pg/ml, TNF – 3.7 pg/ml, and for IL-12p70 – 1.9 pg/ml. We added this information in the Method section (page 10). As an example, cytokine detection ELISA kits by Acro Biosystems has the limitation of detectionable dose set as 5.309 pg/mL (https://www.acrobiosystems.com/A1781-Cytokine-detection-ELISA-kits.html?gclid=Cj0KCQiAtICdBhCLARIsALUBFcGFt-ETCJ48MtzQeu8a5HFrl41FqEBjsGaLubmYXYpZdMTqs7_ugFkaArP3EALw_wcB). Another available ELISA kits from Arigo Biolaboratories has their detection range limit set for IL1 beta – 23.44 pg/ml, IL6 – 7.82 pg/ml, IL8 – 25 pg/ml, IFN-gamma – 10.94 pg/ml and TNF-alpha – 17.19 pg/ml (https://www.arigobio.com/Human-Inflammatory-Cytokine-Multiplex-ELISA-Kit-IL1-alpha-IL1-beta-IL6-IL8-GM-CSF-IFN-gamma-MCAF-and-TNF-alpha-ARG80929.html). Therefore, the threshold values of ELISA test and our CBA kit do not differ enough to be considered as a more valuable alternative. Moreover, the CBA kit allows the determination of several cytokines in one serum sample.
"It is interesting, whether the concentration of the tested cytokines correlated with the immunological parameters? Perhaps, this would explain the better observed lymphocyte profile, and the importance of obtained the results in depression. Especially since not only chronic inflammation, but perhaps, an impaired anti-inflammatory response may underlie depression?"
We have found some correlations between percentages of T-cell subpopulations and serum cytokine levels in TRD patients (Tab. 4). There was a significant positive correlation between IL-10, CD3+CD8+ and CD3+CD8+CD95+ cells, a negative between CD3+CD4+ cells and IL-10 and between CD4+/CD8+ ratio and IL-10. There was a positive correlation between CD3+CD8+CD95+ cells and IL-6. IL-1β concentration was positively correlated with CD3+CD4+CD25+ and CD3+CD4+CD69+ cells. IL-8 levels correlated positively with CD3+CD4+CD69+ cells and CD3+CD4+HLA-DR+ cells. We added this to the Discussion as well (page 9).
Reviewer 2 Report
· The abstract should state briefly the purpose of the research, the principal results and major conclusions. An abstract is often presented separately from the article, so it must be able to stand alone. This section isn't clear. Authors just collecting some ideas. Please, try to improve this section by highlighting the research gap and the novelty of this work. Also, try to lead the reader smoothly to your point.
· The necessity and innovation of the article should be presented to the introduction.
Authors should try to summarize the introduction section in to 3-4 paragraphs maximum.
A flowchart should be added to the article to show the research methodology
· It is suggested to compare the results of the present research with some similar studies which is done before.
The Results and Discussion section is devoted, in large, by representing the research out comes' yielded, but a critical and integrated approach of these outcomes has been made, probable at a distinct "synthesis' and cross-cited subsection. In this distinct subsection the key-aspects that determine the outcomes have to be signified into a descriptive manner.
· Authors should support their conclusion from result with references and also compare the results with previous literature.
Author Response
Answers to the second Reviewer. Corrections are marked in green.
"The abstract should state briefly the purpose of the research, the principal results and major conclusions. An abstract is often presented separately from the article, so it must be able to stand alone. This section isn't clear. Authors just collecting some ideas. Please, try to improve this section by highlighting the research gap and the novelty of this work. Also, try to lead the reader smoothly to your point."
We modified the Abstract slightly. Thus, in the begging, we informed about the purpose of the research: "Although there is some evidence for the involvement of cytokines and T cells in treatment-resistant depression (TRD) pathophysiology, the nature of this relationship is not entirely clear. Therefore we compared T cell subpopulations and serum cytokine levels in TRD patients to find relationships between their immunological profile, clinical presentation, and episode severity." Then, we gave information about study groups and methods used: "Blood samples from TRD patients (n=20) and healthy people (n=13) were collected and analyzed with flow cytometry. We analyzed the percentages of helper and cytotoxic T cells with the expression of selected activation markers, including CD28, CD69, CD25, CD95, and HLA-DR. Also, the serum levels of inflammatory cytokines IL12p70, TNF-α, IL-10, IL-6, IL-1β, and IL-8 were determined." Then, the results: "TRD patients had a lower percentage of CD3+CD4+CD25+ and CD3+CD8+CD95+ cells than healthy people. They also had lower serum levels of IL-12p70 and TNF-, while IL-8 levels were significantly higher. Receiver-operating characteristic (ROC) analysis demonstrated that serum IL-8 above 19.55 pg/ml was associated with the likelihood ratio of 10.26 of developing TRD. No connections between the MADRS score and immunological parameters have been found.". Conclusions: "These results show that TRD patients have reduced percentages of T cells expressing activation antigens (CD25 and CD95) and higher serum of pro-inflammatory and chemotactic IL-8. These changes may indicate a reduced activity of the immune system and the important role of IL-8 in maintaining chronic inflammation in the course of depression."
Hopefully, the Reviewer will be satisfied with our proposition of the abstract.
"The necessity and innovation of the article should be presented to the introduction."
As we mentioned at the end of the Introduction, this paper compares main T cell subpopulations and serum cytokine levels in TRD patients to find relationships between patients' immunological profiles, clinical presentation, and episode severity. Unfortunately, the articles described in the Introduction show no consensus on the immunological profile of patients. Although there is some evidence for the involvement of cytokines and T cells in TRD pathophysiology, the nature of this relationship is not entirely clear. Therefore, we analyzed the percentages of helper and cytotoxic T cells with the expression of CD28 antigen due to its role in antigen presentation and selected activation markers, including CD69, CD25, CD95, and HLA-DR. Also, the serum levels of inflammation-related cytokines: IL12p70, TNF-α, IL-10, IL-6, IL-1β, and IL-8 were determined.
"Authors should try to summarize the introduction section in to 3-4 paragraphs maximum."
We shortened the Introduction, as suggested by the Reviewer.
"A flowchart should be added to the article to show the research methodology"
We prepared a flow chart (Fig. 5), as suggested by the Reviewer.
"It is suggested to compare the results of the present research with some similar studies which is done before."
As we mentioned in the Discussion section, the lack of newer studies makes it impossible to compare the obtained results, especially concerning T-cell subpopulations with the expression of activation antigens. We were able to compare our results concerning cytokine levels and basic T cell subpopulations with other studies (pages 7-8). The results concerning basic T-cell subpopulations (CD3+CD4+ or CD3+CD8+ cells) are in line with more recent studies [Grosse L et al. Psychopharmacology (Berl). 2016 May;233(9):1679-88; Başterzi AD et al. Prog Neuropsy-chopharmacol Biol Psychiatry. 2010 Feb 1;34(1):70-5] (page 8).
"The Results and Discussion section is devoted, in large, by representing the research out comes' yielded, but a critical and integrated approach of these outcomes has been made, probable at a distinct "synthesis' and cross-cited subsection. In this distinct subsection the key-aspects that determine the outcomes have to be signified into a descriptive manner."
The result should present the research outcomes. In the Discussion, we compared our results with similar studies and tried to explain differences in some results. We are not entirely sure what the Reviewer had in mind by writing: "In this distinct subsection the key-aspects that determine the outcomes have to be signified into a descriptive manner." At the end of the discussion, we added a paragraph about the study's limitations (page 9).
"Authors should support their conclusion from result with references and also compare the results with previous literature."
We followed Reviewer's suggestion; we compared our results with some other studies, searched for similarities, looked for differences, and tried to explain them, for example, in paragraphs 2-4 in the Discussion (pages 7-8).
Reviewer 3 Report
In the manuscript entitled “Changes in T cell subpopulations and cytokine levels in patients with treatment-resistant depression”, Szalach et al. characterized the circulating T-cell and serum cytokine profiles in patients who have been clinically diagnosed with treatment-resistant depression. Overall, the manuscript is reasonably well written, with a comprehensive review of current literature on the immune profile of patients with certain psychological disorders. There are some concerns to be addressed before the reviewer can recommend for publication.
Despite the comprehensive review of literature, the reviewer had some difficulty in fully appreciating as to why only the selected T cell subsets and selective cytokines were characterized. Were the authors trying to interrogate certain activation and/or functional pathways? What about the other circulating immune cells, such as B cells, NK cells, monocytes, etc? In addition, within the selected T cell subsets characterized, the reviewer failed to appreciate how these subsets correlated with each other? As an example, with the diminished CD3+CD8+CD95+ subset in the TRD patients but not the other subsets, what did the results imply, in terms of the overall T cells’ activation and function, etc?
For the elevated levels of certain cytokines (e.g. IL-8, TNF, etc) and reduced % of certain T cell subsets (e.g. CD3/4/25) in the TRD patients, were the authors trying to suggest that the observed differences were the cause of TRD? How would the authors exclude the possibility of drug-induced changes? Has any of the medications or treatment regimens as described in Table 1 been implicated in altering the patients’ immune profile? It is important to clarify and provide support on whether the immune profile alterations were the cause or the effect of TRD. Due to the rather small sample size, the authors must also be careful on making statements that IL-8 has a good diagnostic value for TRD and sufficient evidence.
Why was BMI not available on the healthy controls? This information is critical for supporting the proposed correlations and statements between obesity, MDD and MADRS scores (line 219-230). Also, were there any statistical significance (i.e. sampling bias) between the studied cohorts with regards to their sex, age and BMI? The authors need to provide the p values to show that if there were or weren’t.
While the authors were able to show statistically significant correlations between the % of certain specific T subsets and episode length, the correlations were rather poor with the highest r of merely 0.62 (or r^2 of 0.38). The authors must be cautious in making any statements on what is correlating with what based off on these low r values.
The acronyms and abbreviations could use better management and organization. Some were repeatedly defined, while some were defined but not used again. For example, MDD was repeatedly spelled out on line 27 and 33, and TRD was spelled out on line 9, 87, 89, 97, while HPA and ATRQ were defined on line 36 and 273, respectively, but never used again. To improve the read of the manuscript, the authors need to thoroughly proofread to ensure that all acronyms were spelled on their first use, while eliminating the acronyms that were used only once.
In Table 1, most of the acronyms were not necessary as they were only used once, i.e. SSRI, SNRI, SARI, NaSSA, TCA. The authors should consider just spelling them out to simplify the footnote. On the other hand, in all the tables and figures, all acronyms repeatedly used should be spelled out in the footnote, e.g. HC and TRD.
Other typographical errors included:
Line 222 – MBI?
Line 265 – “ . Patients”?
Line 286 – missing the degree symbol
Line 288 – Samples of 100?
Author Response
Answers to the third Reviewer. Corrections are marked in the text in blue.
"In the manuscript entitled "Changes in T cell subpopulations and cytokine levels in patients with treatment-resistant depression", Szalach et al. characterized the circulating T-cell and serum cytokine profiles in patients who have been clinically diagnosed with treatment-resistant depression. Overall, the manuscript is reasonably well written, with a comprehensive review of current literature on the immune profile of patients with certain psychological disorders. There are some concerns to be addressed before the reviewer can recommend for publication."
Thank you for your comments and suggestions.
"Despite the comprehensive review of literature, the reviewer had some difficulty in fully appreciating as to why only the selected T cell subsets and selective cytokines were characterized. Were the authors trying to interrogate certain activation and/or functional pathways? What about the other circulating immune cells, such as B cells, NK cells, monocytes, etc? In addition, within the selected T cell subsets characterized, the reviewer failed to appreciate how these subsets correlated with each other? As an example, with the diminished CD3+CD8+CD95+ subset in the TRD patients but not the other subsets, what did the results imply, in terms of the overall T cells' activation and function, etc?"
Unfortunately, we did not test other populations during the study, as we focused mainly on T cells, especially CD4-positive cells, due to their diverse role in regulating the immune response. In the future, we may also study other populations, such as monocytes or natural killer cells.
As we wrote in the Discussion section, the lack of newer studies makes it impossible to compare the obtained results. However, since CD95 is also associated with the activation of T cells, its deficiency and the deficiency of CD25-positive cells could be associated with disorders of T cell activation in TRD patients. Also, CD25-positive cell deficiency may be associated with an inadequate T-cell response expressed by changes in serum IL-12p70 or TNF-α in TRD patients. Another explanation of CD25 expression in CD4-positive cells is a deficiency in Tregs, which we suggested in the text (pages 8-9).
The lack of change in expression of the CD69 antigen is probably because it is a classical early activation marker that decreases rapidly 4-6 hours after T-cell contact with the antigen. In chronic conditions, CD69 is expressed on infiltrated leukocytes at inflammatory sites rather than in blood. HLA-DR is a molecule typically expressed by antigen-presenting cells (APCs) and is associated with antigen presentation. However, it is also expressed in T cells during their activation. According to some authors, reduced HLA-DR expression is a mechanism adaptive to an overwhelming burst of inflammatory stimuli; high frequency of HLA-DR+ T cells strongly correlates mainly with severe lymphopenia, systemic inflammation, and cytokine storm.
For the elevated levels of certain cytokines (e.g. IL-8, TNF, etc) and reduced % of certain T cell subsets (e.g. CD3/4/25) in the TRD patients, were the authors trying to suggest that the observed differences were the cause of TRD? How would the authors exclude the possibility of drug-induced changes? Has any of the medications or treatment regimens as described in Table 1 been implicated in altering the patients' immune profile? It is important to clarify and provide support on whether the immune profile alterations were the cause or the effect of TRD. Due to the rather small sample size, the authors must also be careful on making statements that IL-8 has a good diagnostic value for TRD and sufficient evidence.
We already made some suggestions about our results in the Discussion section (marked in yellow and blue). We were thinking about drug influence. Unfortunately, TRD patients usually take more than one drug in different combinations. Treating patients with only one type of drug would be essential to compare drug influence. We added this information in the last paragraph of the Discussion about the study's limitations.
"Why was BMI not available on the healthy controls? This information is critical for supporting the proposed correlations and statements between obesity, MDD and MADRS scores (line 219-230). Also, were there any statistical significance (i.e. sampling bias) between the studied cohorts with regards to their sex, age and BMI? The authors need to provide the p values to show that if there were or weren't."
We added information about the BMI of healthy people. There was no difference between TRD patients and HC regarding age or BMI. We added this information in Table 1, along with p values.
"While the authors were able to show statistically significant correlations between the % of certain specific T subsets and episode length, the correlations were rather poor with the highest r of merely 0.62 (or r^2 of 0.38). The authors must be cautious in making any statements on what is correlating with what based off on these low r values."
First of all, we used a two-tailed Spearman correlation because we did not specify the anticipated sign of the correlation coefficient before collecting the data. Also, there was still a significant interaction between these variables after adjusting for potential confounders (sex, age, BMI, and smoking), which could influence our results. We added this information to the results (page 6).
"The acronyms and abbreviations could use better management and organization. Some were repeatedly defined, while some were defined but not used again. For example, MDD was repeatedly spelled out on line 27 and 33, and TRD was spelled out on line 9, 87, 89, 97, while HPA and ATRQ were defined on line 36 and 273, respectively, but never used again. To improve the read of the manuscript, the authors need to thoroughly proofread to ensure that all acronyms were spelled on their first use, while eliminating the acronyms that were used only once."
Thank you for pointing out the incoherence of the organization of acronyms and abbreviations. We applied the suggestions listed above to the text.
"In Table 1, most of the acronyms were not necessary as they were only used once, i.e. SSRI, SNRI, SARI, NaSSA, TCA. The authors should consider just spelling them out to simplify the footnote. On the other hand, in all the tables and figures, all acronyms repeatedly used should be spelled out in the footnote, e.g. HC and TRD."
We have used the acronyms of medications to simplify the table and make it easier to read. The text used acronyms such as HC and TRD several times. We also added it in Table 1. Therefore, we do not need to spell them out in every footnote.
"Other typographical errors included:
Line 222 – MBI?"
We corrected the mistake.
"Line 265 – " . Patients"
The subtitle "Patients" was changed to "Study groups."
"Line 286 – missing the degree symbol"
We put the symbol of the degree.
"Line 288 – Samples of 100?"
We corrected the mistype; the correct unit is microliter (ul).
Round 2
Reviewer 1 Report
Thank you for the corrections, the work in this form can be published.
Reviewer 2 Report
No further revision needed.
Reviewer 3 Report
All previous concerns have been appropriately addressed.